# PeerJ

# There and back again: putting the vectorial movement planning hypothesis to a critical test

Eva-Maria Kobak[1,3] and Simone Cardoso de Oliveira[1,2]

[1] Bernstein Center Freiburg, University of Freiburg, Germany
[2] BrainLinks-BrainTools, Cluster of Excellence, University of Freiburg, Germany
[3] Department of Bioengineering, Imperial College London, United Kingdom

## ABSTRACT

Based on psychophysical evidence about how learning of visuomotor transformation generalizes, it has been suggested that movements are planned on the basis of movement direction and magnitude, i.e., the vector connecting movement origin and targets. This notion is also known under the term "vectorial planning hypothesis". Previous psychophysical studies, however, have included separate areas of the workspace for training movements and testing the learning. This study eliminates this confounding factor by investigating the transfer of learning from forward to backward movements in a center-out-and-back task, in which the workspace for both movements is completely identical. Visual feedback allowed for learning only during movements towards the target (forward movements) and not while moving back to the origin (backward movements). When subjects learned the visuomotor rotation in forward movements, initial directional errors in backward movements also decreased to some degree. This learning effect in backward movements occurred predominantly when backward movements featured the same movement directions as the ones trained in forward movements (i.e., when opposite targets were presented). This suggests that learning was transferred in a direction specific way, supporting the notion that movement direction is the most prominent parameter used for motor planning.

Corresponding author
Eva-Maria Kobak,
eva-maria.kobak10
@alumni.imperial.ac.uk

## INTRODUCTION

An approach frequently used to investigate the functional organization of the motor system is to study learning of visuomotor transformations and the transfer of such learning to untrained conditions. For example, visuomotor rotations have frequently been applied while subjects perform simple reaching movements. Initially, the transformation leads to movement errors reflecting the magnitude of the rotation, but gradually, most subjects are able to adapt to the rotation. For this to happen, the internal representation of the movement has to be changed (*Imamizu, Uno & Kawato, 1995*). There are several ways how this may be achieved: remapping of the locations of origins and targets ("position

remapping") is one possibility. Another is that subjects remember the posture assumed when successfully reaching the target, based on the idea that movements are planned by converging to a final end posture (*Polit & Bizzi, 1978*; *Rosenbaum et al., 1995*). A third possibility is that the alteration of movement direction to a given target is remembered, based on the idea that movements are planned on the basis of the vector connecting the starting location to the target ("vectorial planning", *Gordon, Ghilardi & Ghez, 1994*).

While some early studies provided evidence supporting the idea of final end posture being assumed by subjects (*Polit & Bizzi, 1978*; *Rosenbaum, Meulenbroek & Vaughan, 1999*), other evidence points more towards the vectorial planning hypothesis (*Gordon, Ghilardi & Ghez, 1994*; *Vindras et al., 1998*; *Messier & Kalaska, 1999*; *Krakauer et al., 2000*; *Krakauer, 2009*).

When visuomotor rotations were trained, learning was found to be transferred only to movements in the same direction as the trained one, and not to previously trained targets that were approached from other directions (*Krakauer et al., 2000*; *Wang & Sainburg, 2005*). These studies, however, used different areas of the workspace for learning and test trials, and therefore introduced space as a confounding variable to the experiments. In the *Krakauer et al. (2000)* study, test trials started at the same origin, but were aimed at targets that had not been visited during learning. In the *Wang & Sainburg (2005)* study, the starting positions for test trials were in completely different regions of the workspace, which were not visited during learning at all. The transfer of learning that both studies found for test targets might as well be due to a position remapping learning effect that remained restricted to the trained workspace as to a transfer of the learned direction.

To test this possibility, we applied a revised version of the center-out-and-back task, making use of the fact that backward movements occur in exactly the same location of the workspace as forward movements. Our task provided visual feedback about hand position only during forward movements. During backward movements, the cursor denoting hand position was not visible. Thus, learning could occur only in forward movements, and backward movements could be used for testing of learning transfer. This experiment constitutes an essential test of the vectorial planning hypothesis, the important factor being that position in workspace is not added as a confounding variable (see Fig. 1).

Forward and backward movements in our experiment were separated by a short break at the target, rendering forward and backward movements clearly distinguishable. Although *Krakauer et al. (2000)* also used a center-out-and-back paradigm in their experiments, they did not include any detailed analysis of backward movements. This may partly be due to the fact that their task did not require subjects to stop at the target, making backward movements not very well distinguishable from forward movements. Also, in their task, visual feedback was present throughout the task.

To put the hypothesis of vectorial planning to a crucial test, we designed three different variants of the center-out-and-back paradigm. In all variants, a 60 degrees visuomotor rotation was applied in the learning trials. In the first variant (paradigm one = P1), subjects moved to and from 12 targets distributed on a circle around the central starting position, such that learning during forward movements included movement directions

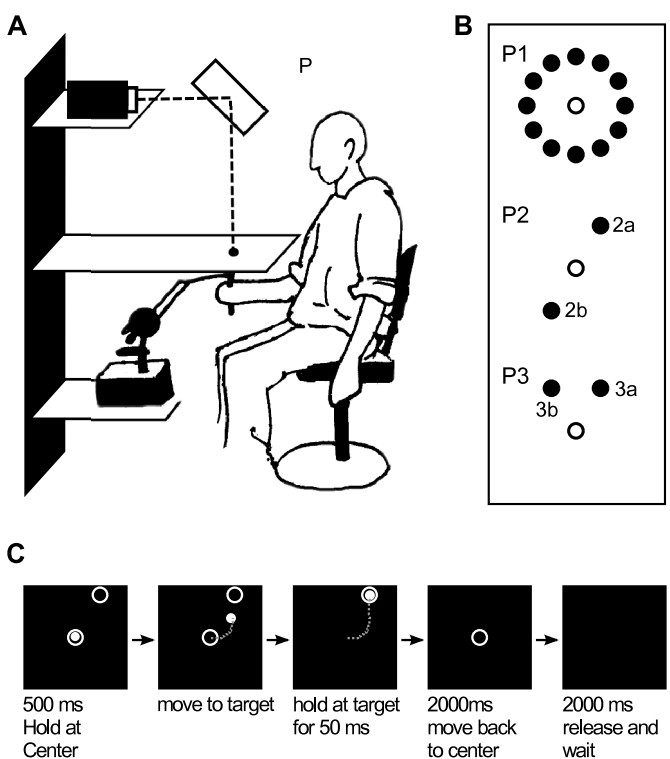

**Figure 1 Experimental setup.** (A) Subject sitting in front of the setup. (B) Target locations in the three paradigms (unfilled circle represents the starting location). (C) Trial sequence. The phantom device was programmed to autonomously move back to the location of the origin during the last 2000 ms while subjects were instructed to release the handle and wait for reappearance of the origin.

from 0 to 360 degrees in 30 degree increments. In the second (paradigm two = P2) and third (paradigm three = P3) variant, movements were directed only to two targets, either 180 (P2) or 60 degrees (P3) apart from each other. This means that in P2, the directions of forward movements to one target were equal to backward movements from the other target, while in P3, backward and forward movement directions did not match.

If subjects would learn the shifted locations of the target and would make use of this knowledge when planning backward movements, transfer of learning from forward to backward movements would occur in all three variants of the paradigm, supporting the position remapping theory. If learning would be based on learning the rotated directions, however, backward movements should only be affected if they would occur in the directions that were trained during forward movements, i.e., in experiments one and two. This would be in accordance with the vectorial movement planning hypothesis (Fig. 2).

We found that transfer of learning from forward to backward movements preferentially occurred in paradigms one and two. Thus, our study further supports the vectorial planning hypothesis and strengthens the idea that movement direction rather than the locations of origin and target is the most prominent parameter used by the motor system for planning movements.

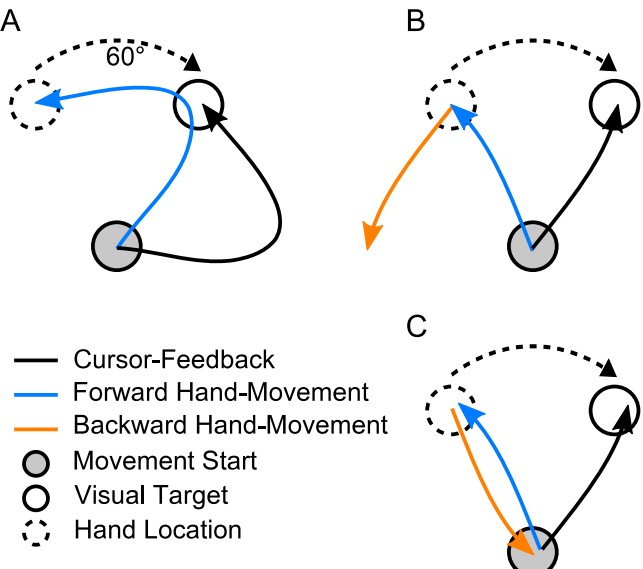

**Figure 2 Expected trajectories in our task.** (A) Initial forward movement under 60 degrees clockwise (CW) visuomotor rotation (beginning of training). (B) Expected forward movement after learning the transformation and expected backward movement based on visual information of target position. (C) Expected forward movement after learning the transformation and expected backward movement, based on hand position.

## MATERIALS AND METHODS

### Experimental setup

A phantom device 1.5 HF (SensAble Technologies, Woburn, MA, USA) was used to track subjects' movements. The resulting trajectories are shown in Fig. 3. The device was programmed to move frictionlessly in a horizontal plane directly under a horizontal board in front of which subjects were seated. During the experiment, the momentary position of the Phantom handle end point was recorded, digitized, visualized on a computer monitor and projected onto the board in such a way that projection and actual position of the phantom end-point were vertically aligned (in the non-rotated condition, see Fig. 1A). The sampling rate for the position recording was 100 Hz, and the gain with respect to the real movements was set to one, i.e., cursor movements had the same amplitudes as hand movements. The phantom device was programmed to autonomously move back to the centre of the horizontal workspace after each trial.

### Participants

All subjects participating in the experiment were right-handed (verified by a modified Edinburgh Handedness Inventory, *Oldfield, 1971*) and naive as to the purpose of the study. Experimental procedures were approved by the local ethics committee (University of Freiburg), and all participants gave their informed written consent prior to starting the experiment. In total, 42 subjects were tested in the experiment. Thirteen of these had to be excluded from analysis since they failed to learn the visuomotor rotation. Successful

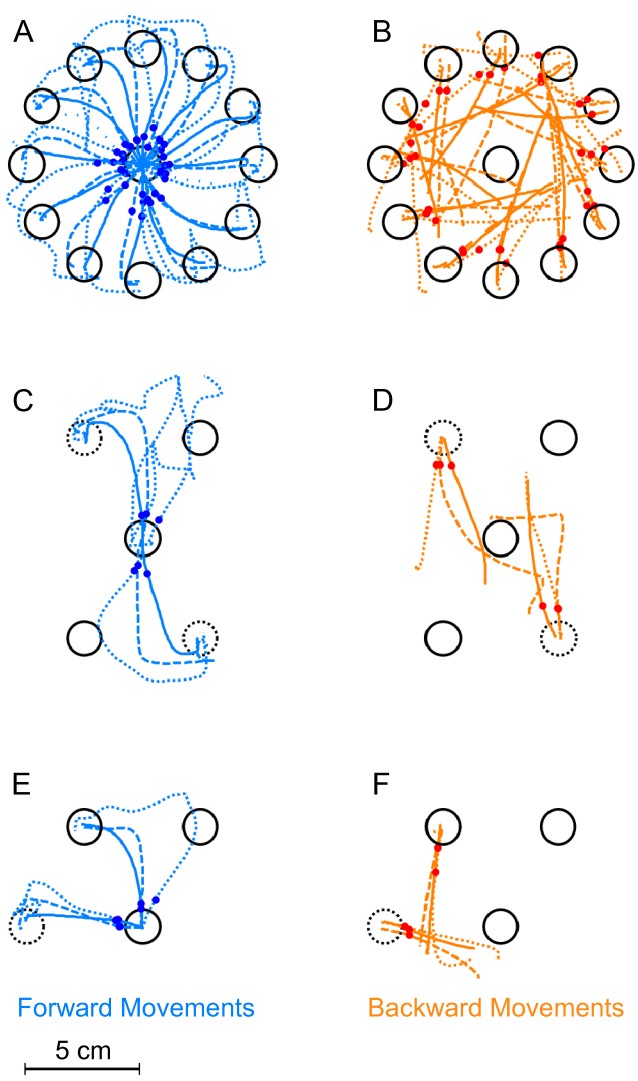

Forward Movements    Backward Movements

5 cm

**Figure 3 Exemplary trajectories during different stages of learning.** Trajectories are shown for one representative subject from each of the three experimental groups. Dotted lines denote the first successful trial, dashed lines a trial during learning (the 8th trial) and the solid lines one of the last successful trials in the block (the 17th in the first experiment, the 173rd for the second and third). (A) Forward movements in paradigm 1. (B) Backward movements in paradigm 1. (C) Forward movements in paradigm 2. (D) Backward movements in paradigm 2. (E) Forward movements in paradigm 3. (F) Backward movements in paradigm 3. The red and blue dots show at what moment initial movement errors were detected in the respective trials (see methods). Locations of the rotated targets are given by the dashed target-circles. All trajectories are shown in hand-space.

learning was defined as a significant difference between the first and last 50 movement errors under the rotation (irrespective of target location, see below), using a standard $t$-test with $p < 0.05$. As we wanted to test the transfer of learning from forward to backward movements, it was necessary to exclude subjects who did not show any learning in the forward movements. Of the remaining subjects, 9 were tested in the first, and 10 each in the second and third paradigm described in the following paragraph (see also Fig. 1B).

## Paradigms and trial-sequence

All subjects participating in our experiment were assigned to one of three experimental groups, each of which was tested in a different paradigm. Subjects in all three groups had to complete three experimental blocks in which they had to perform out-and-back reaching movements. Importantly, visual feedback for their movements was only given during forward movements and switched off when subjects returned to the origin. In the first block (familiarization block, consisting of 120 trials), subjects got veridical visual feedback about their movements. This block was needed to verify that, given non-rotated visual feedback during forward movements, subjects would move straight to the target and back to origin, even if during backward movements, visual feedback was absent. In the second block (learning block, consisting of 360 trials), a visuomotor rotation (60 degrees around the origin of movements) was introduced during forward movements. In the third block (washout block, consisting of 120 trials), veridical, non-rotated cursor feedback was given again during the forward movements.

The three paradigms were set up as follows: the first group of subjects was tested in a paradigm in which one of 12 targets was presented in each trial (P1—targets were chosen pseudo-randomly over each block). The 12 targets were equidistantly distributed (i.e., 30 degrees apart from each other) on a circle with radius 5 cm around the movement origin (Fig. 1B, P1). The second group of subjects was tested in a paradigm in which one of only two targets was presented in each trial (P2). The two targets were positioned at 60 and 240 degrees from the origin (0 degrees meaning rightward, 90 degrees meaning forward direction, seen from the perspective of participants, see Fig. 1B, P2), i.e., at 180 degrees from each other. In the figures depicting the targets separately, these targets will be referred to as P2a (at 60 degrees) and P2b (at 240 degrees) (see Figs. 8–10). The third group of subjects was also presented with one of two targets in each trial, but the target locations at 60 and 120 degrees from the origin were only separated by 60 degrees, (see Fig. 1B, P3). Analogous to the previous paradigm, these targets will be referred to as P3a (at 60 degrees) and P3b (at 120 degrees) (see Figs. 8–10).

It is important to note that since the target number differed between paradigms but the trial number in each experimental block was kept constant, the number of repetitions to each target differed. In the first paradigm, each target was shown 10 times in the familiarization and washout blocks, and 30 times in the learning block. In the second and third paradigm, each target was presented 60 times during familiarization and washout and 180 times during learning.

In the beginning of each trial, subjects had to position the cursor in the centre of the workspace, which was indicated by a circle 1.5 cm in diameter. After an initial 500 ms waiting-period (Fig. 1C), the target (another circle 1.5 cm in diameter) appeared, and subjects had to reach it with the cursor. At the same time, the circle showing the centre of the origin disappeared. Only after the cursor was placed within the target and remained there for 50 ms, subjects were allowed to move back to the origin. This was signified by the disappearance of the target and the cursor, and the reappearance of the circle representing the origin. Prior to starting the experiment, subjects were instructed to move back to the centre

of workspace as accurately as possible, albeit lacking visual feedback about the cursor position. With regards to the speed of movements, subjects were told to move at a natural pace. After they thought they had reached the origin, subjects were requested to let loose of the phantom handle and put their hand on their knee until the next trial started. During this period, the handle was autonomously moved back to the exact origin position by the phantom device. Since subjects were not allowed to keep the phantom handle in their hand, they could not feel whether or in which direction the phantom was moving to get back to the centre, and therefore they had no proprioceptive information on whether or not they had actually reached the origin during the backward movements. One second after the Phantom device was reset to the central position, the central target and the cursor reappeared, signalling the start of the next trial and instructing subjects to grasp the handle again.

## Data analysis

Before analysing the movements, we low-pass filtered the recorded trajectories (10 Hz cut-off, 2nd order Butterworth filter). Oscillations above 10 Hz are unlikely to be caused by subjects' movements, and are therefore assumed to represent recording artefacts. For quantification of subjects' learning, we determined the error of initial movement direction. This error was defined as the signed angular difference between initial movement direction and target direction from the hand location at movement onset. Initial movement direction was defined as the hand position 150 ms into the movement in relation to the hand location at movement onset. Movement onset was found by a semi-automated procedure, aimed at determining a point in time when the handle had not been moved from the starting location. At the same time, we wanted to exclude quivering movements as well as 'false starts' from the trials entering the analysis. To this end, we defined movement onset as the point in time 100 ms before 45 percent of maximal hand-velocity was reached. In case velocity was not increasing monotonically until the threshold of 45 percent maximal velocity was reached, the procedure was repeated for the next time point (and monotony tested for 100 ms after that new time point) until the velocity increase was monotonic. After a movement onset was found by the automated procedure, we visually inspected its relation to both, movement trajectory and speed-profile of the movement. If either no movement onset was found, or it was obviously defective, we discarded the respective trial. Initial movement direction was chosen as parameter for assessing the subjects' behaviour because we wanted to assess the change in the internal model of the subjects used for movement planning, before any online corrections, induced by the visual feedback, took place. Although, in the literature, latencies of under 150 ms have been described for visual feedback influencing motor control (*Franklin & Wolpert, 2008*), in our task, we did not observe any corrective movements before 150 ms after movement onset (see Fig. 4 for an exemplary movement, with the portion of the trajectory used for determining the initial movement direction highlighted in colour).

To show the time course of performance and learning in the experimental blocks, we plotted initial movement errors against trial number. The time course of initial movement errors was fit with a linear function in the familiarization blocks, and with a single

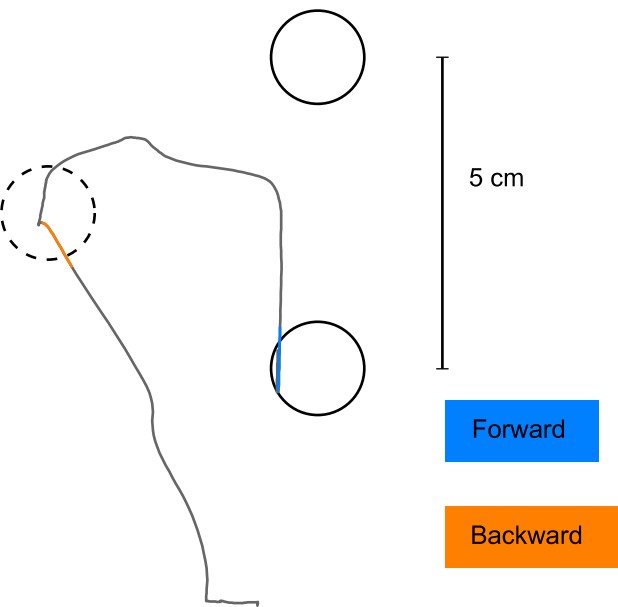

**Figure 4 Detection of initial movement error.** The coloured parts of the trajectories show the interval of the movement which was used to define the initial movement error (in blue for the forward movement, in orange for the backward movement). Note that the corrective movement in forward direction starts well after the point in time when initial movement error was detected.

**Table 1 Parameters of curves fit to initial movement errors.**

| Paradigm | Figure | target | a | b | $\tau$ | MSE |
|---|---|---|---|---|---|---|
| P1 forward | 3 | All | 8.3–11.9 | 33.4–38.5 | 81.5–120.3 | 276 |
| P1 backward | 3 | All | 12.9–14.4 | 17.8–30.0 | 15.4–31.8 | 281 |
| P2a forward | 4A | 60° | 15.5–18.2 | 30.2–38.9 | 23.9–38.1 | 262 |
| P2a backward | 4A | 60° | 18.4–20.2 | 18.9–36.1 | 5.3–13.2 | 257 |
| P2b forward | 4B | 240° | 14.2–17.9 | 29.2–35.6 | 34.5–56.6 | 216 |
| P2b backward | 4B | 240° | 15.4–17.5 | 10.6–52.2 | 0.8–6.7 | 397 |
| P3a forward | 5A | 60° | 8.8–10.8 | 36.2–45.4 | 19.1–27.3 | 207 |
| P3a backward | 5A | 60° | 23.5–26.3 | 7.6–18.4 | 8.2–46.6 | 350 |
| P3b forward | 5B | 120° | 7.7–9.6 | 21.5–29.3 | 18.1–30.5 | 179 |
| P3b backward | 5B | 120° | 14.5–15.8 | −3.3–15.5 | −5.3–16.3 | 156 |

**Notes.**

P1, paradigm 1, all 12 targets; P2a, paradigm 2, target at 60°; P2b, paradigm 2, target at 240°; P3a, paradigm 3, target at 60°; P3b, paradigm 3, target at 120°. For each parameter of the fit, the confidence intervals are given; MSE, mean squared error.

exponential function according to $E(t) = a + b \times e^{(-t/T)}$ in the learning and washout blocks. The parameters derived for the exponential fits are shown in Table 1.

For each trial, we tested whether movement errors (pooled over subjects) were significantly different from zero (ranksum test, $p < 0.01$, see Figs. 5–7). This is mostly relevant for the familiarization block, in which initial movement errors from only very few trials are significantly different from zero (not aimed more or less directly in the direction of the target presented).

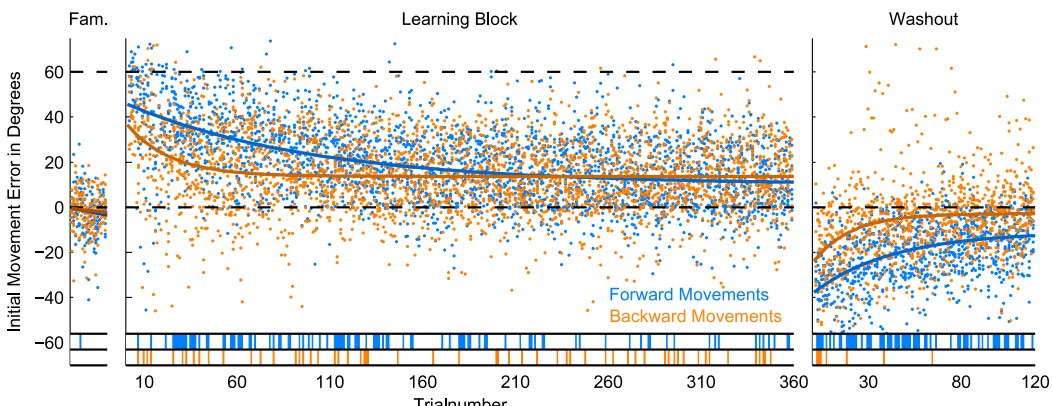

**Figure 5** **Initial Movement Errors (IME) in the first paradigm (P1) with 12 targets.** In the familiarization block (far left) and the washout block (far right), feedback was veridical. In the learning block (middle), visual feedback was rotated around the movement origin by 60 degrees (CW). Individual errors are shown as dots, solid lines denote the exponential fits to the data (for confidence intervals of the fit parameters, refer to Table 1). The bars at the bottom off the graphs denote individual trials in which the errors were significantly different from zero (ranksum test, $p < 0.01$). Forward movements are shown in blue, backward movements in orange.

For Fig. 8, we pooled initial movement errors over a number of trials in the beginning and in the end of the learning block to check for significant differences between forward (Fig. 8A) and backward (Fig. 8B) movements. For the statistical comparison, we used a ranksum test (for the $p$-values, refer to Figs. 8 and 9). For the first paradigm, we pooled over the first and last 50 trials, irrespective of target direction. Target direction was ignored in this paradigm because we expected a high degree of generalization of learning between nearby targets (also see Discussion). For the second and third paradigm, we first sorted trials by target direction and then pooled over the first and last 15 trials separately for each target (P2a and P2b in the second paradigm; P3a and P3b in the third paradigm). Differences between performance in forward and backward movements are shown for the beginning (Fig. 8C) and the end (Fig. 8D) of the learning block. The same analysis for the washout block is shown in Fig. 9.

To directly compare the learning of the rotation in forward versus backward movements between the second and third paradigm, we computed the difference between movement errors in forward and backward direction separately for each subject in the end of the learning block (as for Fig. 8, we averaged over the last 15 trials), and compared these subject-wise differences between the two groups (ranksum test). The individual subject-wise differences are shown in Fig. 10.

## RESULTS & DISCUSSION

### Movement trajectories

Figure 2 schematically illustrates the kinds of forward and backward movements that could be expected under a visuomotor transformation, based on previous findings observed in this kind of task (*Krakauer et al., 2000*; *Krakauer, 2009* for a review on visuomotor rotations).

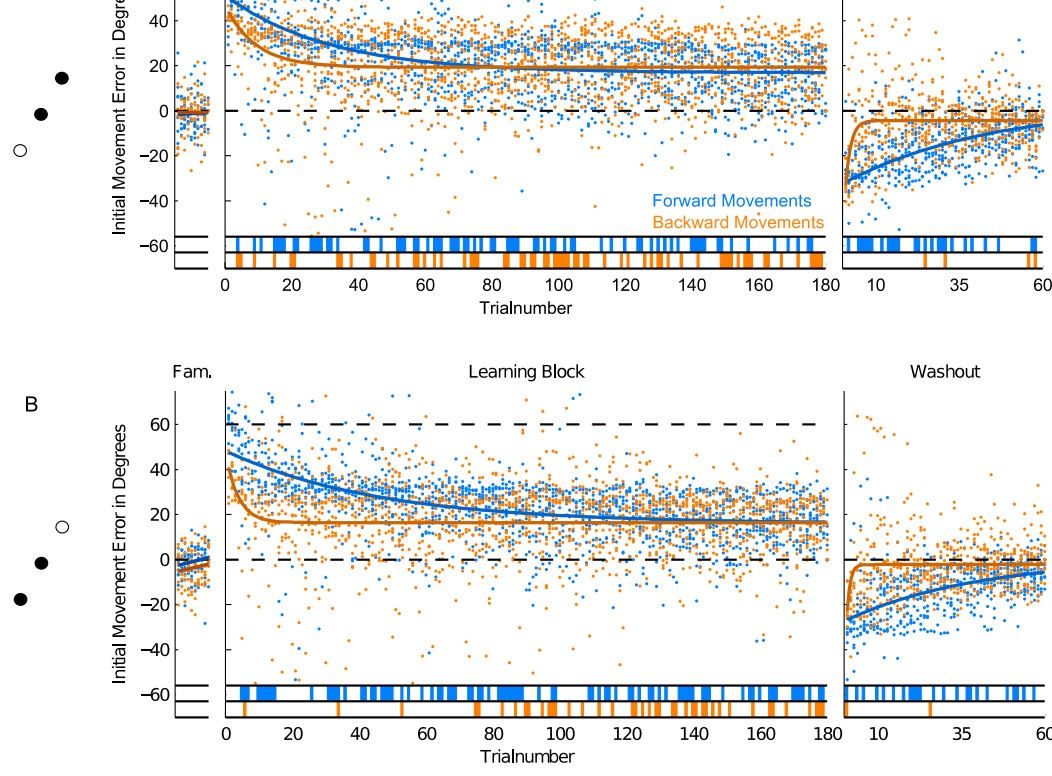

**Figure 6 Initial Movement Errors (IME) in paradigm 2 (P2—targets located 180 degrees apart).** (A) Results for the trials to the target at 60 degrees. (B) Results for the target at 240° in the bottom row of the figure. Refer to Fig. 3 for further details.

Figure 2A (blue arrow) shows the behaviour that can be expected for forward movements in the beginning of exposure to the transformation: subjects would start off moving towards the visually perceived location of the target. After initiating the movement, visual feedback would make them realize that they are 60 degrees off the desired direction, leading to a large corrective movement (resulting in a hook-shaped trajectory). With prolonged exposure to the rotation, subjects would be expected to recalibrate their motor system such that they would immediately reach into the required rotated direction, producing straight trajectories again (Figs. 2B and 2C).

Figure 2B (orange arrow) shows how subjects would move backwards if they would only take the visually perceived target locations into account, failing to account for the shift induced by the transformation, and making them end up in a position completely off the origin. Figure 2C, in contrast, shows how subjects would move back if they would have learned the effect of the transformation, allowing them to faithfully reach the origin again. If such learning transfer to backward movements was seen in all paradigms, it could be concluded that subjects learned a position remapping of all positions visited during learning in forward movements. If, however, learning transfer was observed only in

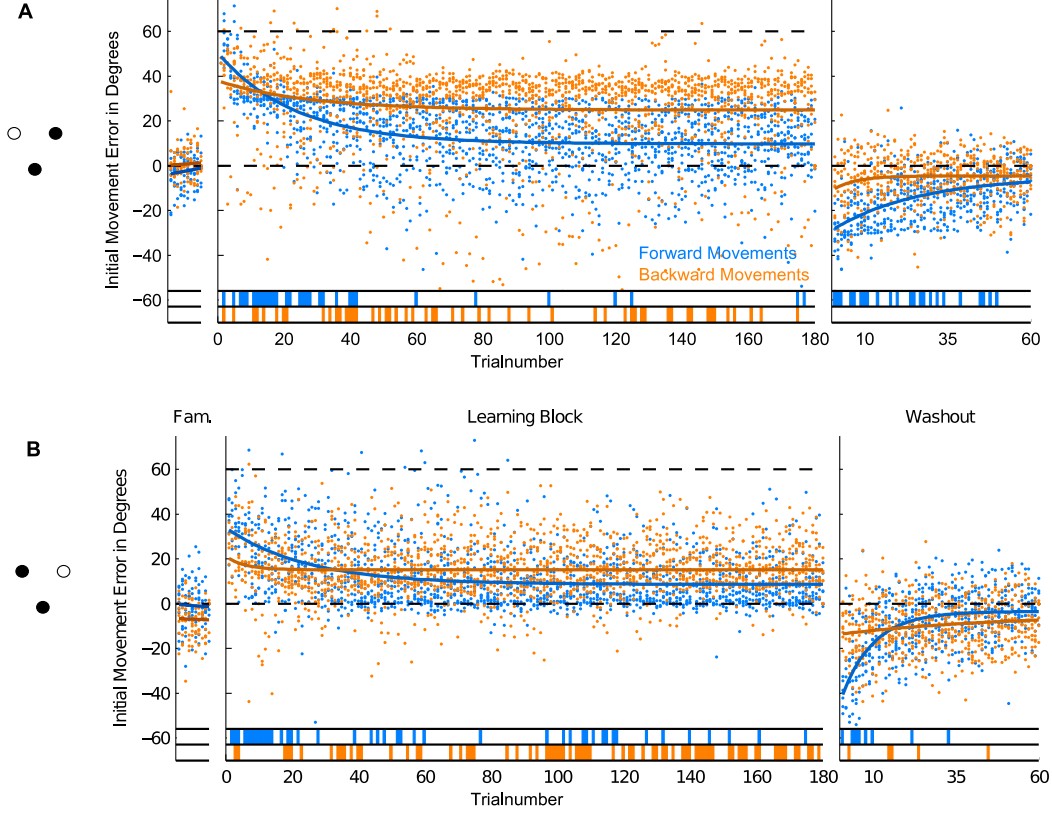

**Figure 7 Initial Movement Errors (IME) in paradigm 3 (P3—targets located 60 degree apart).** Results for the trials to the target at 60 degrees are shown in the top row, results for the target at 120 degrees in the bottom row of the figure. Refer to Fig. 3 for further details.

paradigms one and two, and not in paradigm three, it could be concluded that no general position remapping took place, but rather a specific learning transfer concerning the movement directions trained during forward movement. Note that backward movements are expected to be always straight, since, due to the lack of visual feedback, no corrective movements are expected.

Figure 3 shows typical real examples of movements trajectories in the three paradigms at different stages of the learning block. In the forward movements of all three paradigms (Figs. 3A, 3C and 3E), subjects behaved as expected once the rotation was applied. In the first couple of trials, there were large initial movement errors and subsequently large movement corrections to reach the targets. Over the course of learning, forward movements gradually became straighter and the movement direction rotated more and more to compensate for the rotation, such that the target could successfully be reached. over the course of the learning block.

In contrast to forward movements and as expected due to the lack of feedback, backward movements (Figs. 3B, 3D and 3F) were straight even in the beginning of the learning block. They never displayed the typical hook-like shape induced by corrective movements. Interestingly, backward movements never exactly pointed towards the origin as in Fig. 3C.

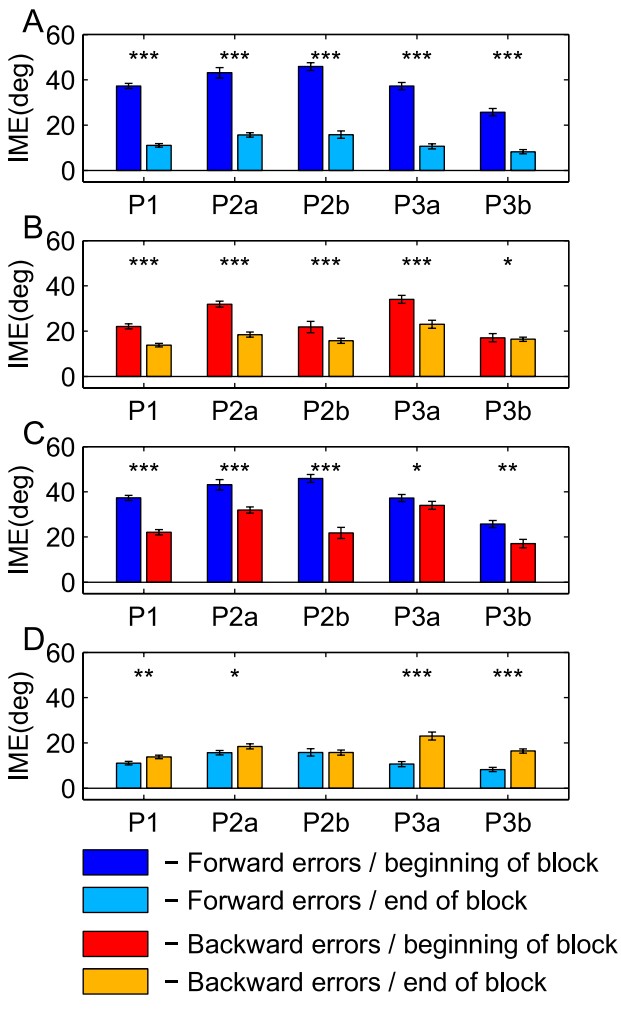

*p<0.05; **p<0.01; ***p<0.001

**Figure 8 Comparison of initial movement errors in the beginning and in the end of learning for forward and backward movements.** (A) Comparing forward movements at the beginning and in the end of the learning block. (B) Comparing backward movements at the beginning and in the end of the learning block. (C) Comparing forward and backward movements at the beginning of the learning block. (D) Comparing forward and backward movements in the end of the learning block. Distributions were tested for significant differences with a ranksum test. P1, paradigm 1, all targets; P2a, paradigm 2, target at 60 degrees; P2b, paradigm 2, target at 240 degree; P3a, paradigm 3, target at 60 degree; P3b, paradigm 3, target at 120 degrees. For paradigm 1, the beginning of the learning block is taken as the first 50 trials, and the end of the learning block as the last 50 trials. In paradigms 2 and 3, beginning means the first 15 trials, and end the last 15 trials. The height of the bars corresponds to the mean over trials. Error bars show the standard error of the mean (SEM) over trials and subjects.

Rather, the initial movement directions seemed to lie between the one required taking the actual hand position into account and the one required based on the visually perceived hand location. In paradigms one and two (Figs. 3B and 3D), it seemed that the backward movement direction would shift over the course of the learning block towards the direction required by the visuomotor rotation.

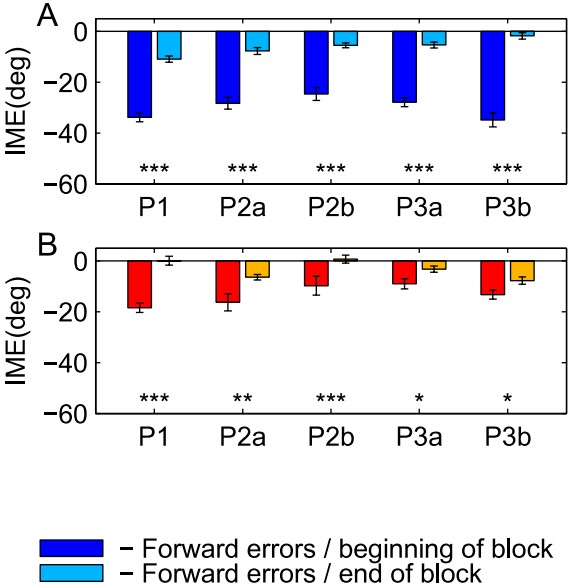

**Figure 9 Differences in performance at the beginning and the end of the washout block.** (A) Forward movements. (B) Backward movements. For color legend and significance levels see Fig. 6.

## Initial movement errors

To quantify motor behaviour and check for systematic changes in movement direction over the course of the experiment, we computed initial errors in forward and backward movement directions and plotted them against time. For better visualization and quantification of the results, we fitted the forward and backward movement errors of the training and washout blocks with an exponential function (see methods for procedure and Table 1 for the estimated parameters of the fits).

For the presentation of initial movement errors in the first paradigm, we ignored target directions when looking at the time-course of learning (see Fig. 5). Due to the pseudo-random presentation of many target directions, consecutive trials to the same target can be separated by many trials to other targets. If targets are learned completely separately, this should not strongly affect the time-course of learning the rotation for single targets (see *Krakauer et al., 2000*), but in our first paradigm, targets were quite close together (30 degrees) so we expected to see at least some generalization between nearby target locations. Instead of trying to correct for this during the analysis (e.g., by taking into account the presentation order of targets), we decided to look at consecutive trials irrespective of target direction for the paradigm with 12 targets.

In the second and third paradigm, target locations were separated by at least 60 degrees (in paradigm three), so we did expect much less, if any, generalization across target directions. In the last two paradigms, we therefore separated trials by target direction and looked at the time-course of learning separately for each of them. This also enabled

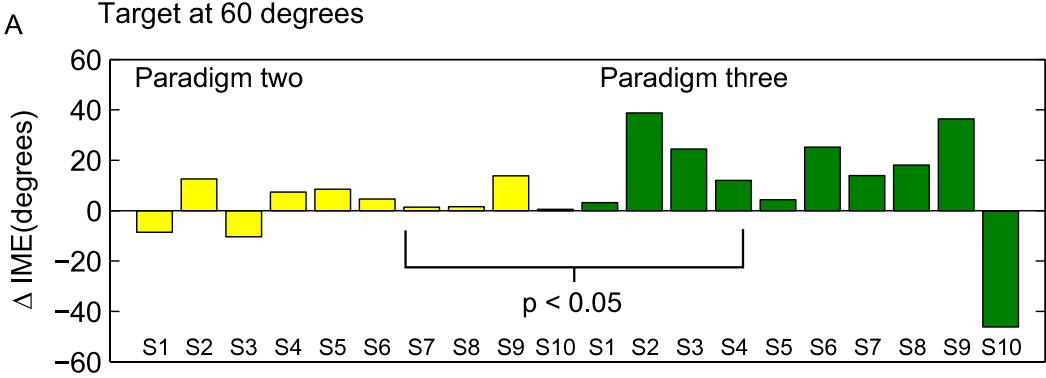

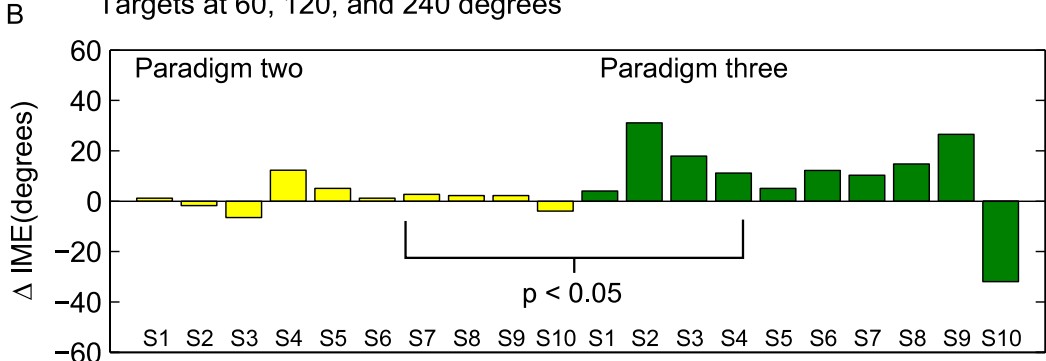

**Figure 10 Differences between forward and backward learning depending on the learned directions.** Each bar shown in the figure represents the difference between initial movement errors in forward and backward direction in the end of the learning block for one subjects. End of the learning block here is equivalent to the last 15 trials in the learning block (compare also methods and Fig. 9). S1–S10 are the 10 subjects in each of the paradigms (note that S1 in the second and the third paradigm was not the same person!). In A, differences are computed only for target P2a (paradigm two) and P3a (paradigm three), respectively. In B, differences are computed over all targets presented in the two paradigms (P2a, P2b, P3a, and P3b). The differences were significantly different between the two groups (ranksum test, $p < 0.05$).

us to compare movements to the same location in the workspace between paradigms: the location of target P2a in the second paradigm is identical to that of target P3a in the third paradigm (see Figs. 6 and 7). The different numbers of targets presented in the paradigms mean that one has to be cautious in comparing learning between the first and the last two paradigms. The conclusions drawn in the following are therefore most critically based on the comparisons of forward and backward movements within a given paradigm and the comparison of paradigm 2 and 3 (in which equal numbers of repetitions were used). Therefore, the factor of target numbers is not confounding our results.

## Forward movements

In forward movements (blue dots and lines in Figs. 5–7), as expected, when the rotation of the visual feedback was switched on, subjects in all groups started off with initial movement errors close to the magnitude of the rotation (60 degrees, see Figs. 5–7, blue

dots). In the following trials, initial movement errors decreased until reaching a plateau at which performance remained relatively constant.

Performance in the washout block (right panel of Figs. 5–7) confirmed that a typical sensorimotor learning process had taken place in forward movements. In all experiments, a distinct after-effect was observed: in the beginning of the washout block subjects started off with large errors in the direction opposite to the transformation. These errors decreased rapidly over the following trials.

In accordance with the learning process, over the course of both the learning and the washout block, the occurrence of significant differences from zero (ranksum test, based on the movements of all subjects in each experimental group for individual trials), seemed to consistently decrease (blue lines in the bottom of Figs. 5–7). Given the low statistical power of comparing a sample of only 10 to a mean of zero, these data have to be interpreted with caution, however.

For the comparison of initial movement errors in the beginning and in the end of the learning block, we pooled over the first and last 50 trials of the block for paradigm one, and over the first and last 15 trials for paradigms two and three. Pooling over a large number of trials will give more reliable results given the variability of initial movement errors, but it will fail to capture the initial phase of the experiment in which the movements are not learned, yet. In the first paradigm, taking 50 trials was reasonable looking at the time-course of learning over consecutive trials (Fig. 5). In the second and third paradigm, especially when looking at targets separately (Figs. 6 and 7), learning is faster, so we chose a smaller number of trials to represent the beginning and the end of learning respectively. Comparing the pooled trials from the beginning and the end of the learning block yielded a highly significant drop of errors in all experiments, showing that substantial learning was achieved in forward movements (Fig. 8A). Note that this finding is obvious to some degree, since subjects who did not show a significant decrease in forward movement errors were not included in the analysis. In the washout block, we found substantial unlearning, again indicated by a significant drop in the initial movement errors from the first to the last trials in the block (Fig. 9A).

Subjects' movement errors in the third paradigm generally seemed to be smaller (both in the beginning and in the end of the learning block) than in the second paradigm (see Fig. 8A). This could, for instance, reflect some generalization of learning between targets (targets were only 60 degrees apart in the third, but 180 degrees apart from each other in the second paradigm), however, further experiments would be needed to investigate this effect in detail.

The learning and unlearning processes seemed to be well captured by the exponential fits (see Table 1 for the parameters of the exponential fits).

## Backward movements

In backward movements (orange dots and lines in Figs. 5–7), the initial movement errors in the very first trials of the learning block were typically smaller than those observed in forward movements (see Fig. 8C). This is in agreement with the observation in movement

trajectories that subjects seemed to plan movements in directions between those required on the basis of the visual and on the basis of the proprioceptive information about the hand's location. Apparently, backward movements were planned by integrating proprioceptive information about the actual hand starting position and the visual information about the cursor position, which is in agreement with other psychophysical experiments (*van Beers, Sittig & Denier van der Gon, 1996*; *van Beers, Sittig & Denier van der Gon, 1999*).

In addition, in backward movements, as in forward movements, errors decreased over the course of the learning and the washout blocks, indicating that the transformation was also learned in backward movements. Since subjects did not get sensory feedback about their performance in backward movements, a learning mechanism depending on the observation of movement errors, like in forward movements, was not possible.

Instead, subjects may have learned over the course of the learning block to change the relative weight they attribute to proprioception as compared to vision when planning their movements. It has been shown that the integration of visual and proprioceptive information used for movement control can be altered by task circumstances (*Touzalin-Chretien, Ehrler & Dufour, 2010*). Generally, subjects tend to rely more heavily on vision than on proprioception for the planning and execution of movements (e.g., *Botvinick & Cohen, 1998*; *Ernst & Banks, 2002*) and proprioceptive information is even suppressed in the beginning of reaching movements (*Shapiro, Gottlieb & Corcos, 2004*; *Shapiro et al., 2009*; *Niu, Corcos & Shapiro, 2012*). During the learning block, subject might learn that the visual feedback they get is unreliable, and chose to disregard it more and more for the planning of the backward movements. Indeed, depending on availability and/or reliability of sensory information, the weighting of proprioceptive and visual information has been found to be subject to change (*Botvinick & Cohen, 1998*; *van Beers, Sittig & Denier van der Gon, 1999*; *Ernst & Banks, 2002*; *Sober & Sabes, 2003*).

Alternatively subjects might learn the shifted target locations during forward movements (consistent with the position remapping hypothesis), and switch to location remapping for planning their backward movements.

Both these learning mechanisms would rely on subjects being able to adopt more than one learning strategy, and switch between them. Indeed, subjects have been reported to readily learn more than one transformation in case they are training in different contexts (*Thomas & Bock, 2012*), and to be able to switch between different control mechanisms depending on that context (*Scheidt & Ghez, 2007*; *Ghez, Scheidt & Heijing, 2007*; *Scheidt, Ghez & Asnani, 2011*). The absence of visual feedback in backward movements could function as a cue to switch context, allowing them either to reweigh the available proprioceptive and visual information or to switch from the control of movement direction when visual feedback is provided to a location remapping mode when it is absent. Very recently, there has been a study proposing that both, the rotated goal location and the rotated direction of movements are learned when subjects are confronted with a visuomotor rotation. This study is in accordance with our results, and also includes a computational model for the learning mechanisms. However, this study—like the ones before—tested the directions in a separate location of the workspace (*Wu & Smith, 2013*).

In our experiment, it was not possible to determine which of the above described mechanisms accounted for the observed learning, or whether maybe both were involved.

Moreover, there is a third mechanism, that we think plays a major role in the learning of the backward movements—the direction-specific transfer of learning from forward to backward movements. In the end of the learning block, performance during backward movements in the first two paradigms almost reached the level observed in forward movements, while in the third paradigm, it was significantly lower (Fig. 8D). Actually, the residual error in the end of learning in paradigm 3 was more than twice as large in backward movements as compared to forward movements.

The difference between the last two paradigms was most striking when comparing the movement errors to the target located at 60 degrees—target P2a in the second, and P3a in the third paradigm. The location of the respective target is the same in both groups (making this comparison as fair as it can get), and the number of repetitions to the target, as well as the number of additional targets presented (one) are the same in the last two paradigms. And yet, in P2a, there was only a slight difference in the movement errors between forward and backward movements in the end of the learning block, whereas in P3a the difference was large and highly significant (Figs. 8D, P2a and P3a, ranksum test, $p$-values see figures). When doing a subject-wise comparison of the difference between forward and backward movement errors between paradigms (see methods for details), we found that the difference was indeed significantly larger in the third paradigm (ranksum test, $p < 0.05$), both when looking only at target P2a/P3a as well as when pooling over all three targets presented in the two paradigms. We propose that this difference is caused by a direction-specific transfer of learning from forward to backward movements. The differences between forward and backward movement errors for single subjects (Fig. 10) reveal that, indeed, the difference between forward and backward movement errors was generally larger in the paradigm in which targets were not located opposite each other. Additionally, this figure shows that there is one subject in paradigm three who acts contrary to any other subject in the group (the last bar on the right in both Figs. 10A and 10B). There was no objective reason to exclude this subject from the analysis, however.

Direction-specific transfer of learning has been described before (*Krakauer et al., 2000*; *Sainburg et al., 2003*), and, in addition, it was shown that learning of visuomotor rotations was transferred between movements starting at different locations of the workspace, as long as they were in the same direction (*Wang & Sainburg, 2005*). In contrast to these earlier studies, however, our study has the advantage that the learning transfer was tested in an area of the workspace that was completely overlapping with the area trained during learning. Therefore, the workspace could be excluded as a confounding variable against location remapping. Our results are consistent with the view that movement direction is a major parameter specified during motor programming, encoded separately from position in the motor system, as postulated by the vectorial planning hypothesis. The idea that movements are primarily planned on the basis of movement direction is also supported by neurophysiological findings that movement direction is a prominent parameter encoded in neuronal activity (*Georgopoulos, Schwartz & Kettner, 1986*). On the basis of

this assumption, planned movements can amazingly accurately and robustly be decoded from neural signals in monkeys (*Schwartz, 1994*). Directional coding of movements seems also to occur in humans (*Cowper-Smith et al., 2010*; *Fabbri, Caramazza & Lingnau, 2010*). Brain-machine interfaces can quite successfully exploit decoding of intended movement directions for steering machines or prostheses with brain signals, both in monkeys (*Taylor, Tillery & Schwartz, 2002*) as well as in humans (*Milekovic et al., 2012*; *Hochberg et al., 2012*).

We propose that the three mechanisms discussed above—the re-weighting of sensory information, the learning of transformed target location, and the (direction-specific) transfer of learning from forward movements—are the most important factors leading to the adaptation to the rotation of movement directions seen in backward movements. However, there are two further possibilities that we want to discuss: first, learning might at least partly have been caused by non-specific generalization of learning (generalization across directions). Although Krakauer et al. in their original study claim that the learning of a visuomotor rotation is local with regards to the direction of movements, their results suggest at least some degree of generalization (*Krakauer et al., 2000*, Figs. 7A and 7B), especially where there is more than one training direction (see also *Brayanov, Press & Smith, 2012*). We think, however, that at 120 degrees, the difference between forward and backward movement directions in paradigm three was large enough to render generalization across them unlikely.

Secondly, subjects might have made use of the mismatch between the location where they left the handle of the tracking device in the end of each trial and the position in which they took hold of it again after the Phantom had moved the handle towards the origin, and started the new trial. To minimize this kind of information, subjects were told to put their hands well away from the tracking device and put them on their lap between trials. Since the movements bringing back the Phantom to the origin were quite small, and given this intermittent movement to the subjects' lap and back, we think this effect is negligible.

While we therefore are confident that we can exclude the latter explanations for the learning in backward movements, further experiments would be necessary to distinguish between the other two explanations mentioned and specify the magnitude to which each adds to the observed learning effect.

To summarize, the main two findings of our study, shedding light on the planning of backward movements in human subjects are: first, when confronted with a transformation at a given moment, the following movement (in our case, the backward movement) is planned based on a mixture of visual and proprioceptive information, suggesting that the transformation induces a re-weighting of information coming from the two perceptual channels. This has been shown before for other types of movements, but to our knowledge, it is a new finding when looking at backward movements specifically. We think that this is relevant because backward movements are an integral part of reaching movements, and it would therefore not be implausible to assume that they are planned along with the forward movements, integrated in one motor command. Our results suggest in contrast, that backward movements are planned separately from the preceding forward movements.

Secondly, we have shown that the learning of a visuomotor rotation in forward directions also induces learning in the following backward movements of out-and-back reaching movements. There are several mechanisms which could be relevant in this respect, and further experiments would be needed to determine their specific contributions to the observed learning effect, but our results strongly suggest that one of them is a direction-specific transfer of learning from forward to backward movements, supporting the vectorial planning hypothesis of motor control and emphasizing the role of direction as an important control parameter within the motor system.

## ACKNOWLEDGEMENTS

The authors would like to thank Carsten Mehring (University of Freiburg) for long-term support and advice and Dmitry Kobak (Champalimaud Neuroscience Programme) for helpful discussions.

### Funding

This project was partially supported by the German Federal Ministry of Education and Research (BMBF grant 01GQ0420 within the Bernstein Centre of Neuroscience, BMBF grant 01GQ0830 to BFNT Freiburg/Tuebingen and within the German-Israeli Project Cooperation: DIP, Project METACOMP), within the Bernstein Center for Computational Neuroscience Freiburg (Grant No. 01GQ0420). EK was also supported by the Department of Bioengineering, Imperial College London. The funders had no role in study design, data collection and analysis, decision to publish, or preparation of the manuscript.

### Grant Disclosures

The following grant information was disclosed by the authors:
German Federal Ministry of Education and Research: BMBF 01GQ0420, 01GQ0830.
Department of Bioengineering, Imperial College London.
German Research Foundation (DFG).
Albert Ludwigs University Freiburg.

### Competing Interests

The authors declare they have no competing interests.

### Author Contributions

- Eva-Maria Kobak conceived and designed the experiments, performed the experiments, analyzed the data, wrote the paper, prepared figures and/or tables, reviewed drafts of the paper.
- Simone Cardoso de Oliveira conceived and designed the experiments, wrote the paper, reviewed drafts of the paper.

## Human Ethics

The following information was supplied relating to ethical approvals (i.e., approving body and any reference numbers):

Ethics Committee of the University of Freiburg: Approval number 187/05.

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
