# Peer review of "There and back again: putting the vectorial movement planning hypothesis to a critical test"

_PeerJ, doi:10.7717/peerj.342_

## Round 0.1 · original submission · Major Revisions

Both external reviewers appreciated your work but raise some points that should be clarified. I consider of particular relevance their doubts on the analysis.

Reviewer 1 ·

Basic reporting

No Comments

Experimental design

The experiment is well designed. The presentation of the results is clear. Somme information about the instructions given to the subjects relative to movement speed is lacking. The authors should report the results of the statistical analysis into the text.
Why Figure 9 is at the end of the paper? Alson into the text it appears after figure 2. Please change.

Validity of the findings

-I think that movement amplitude (5cm) was too short to speak about 'movement'.
-Authors argue about movement direction as a major parameter of movement planning. In generally, this is true. However, in their experiment they manipulated only movement direction, movement amplitude was only 5cm, and they measured only direction errors. Thus, they must formulate more moderate conclusions.
-They argued in favor of the 'vectorial planning hypothesis' because the significance lever, although present, was weaker in P3. Do they think that this is a sufficient argument?
The last conclusion (lines 366-369) is not original.

Reviewer 2 ·

Basic reporting

Figures

In all the figures I suggest to put a legend of the colors. This would simplify and speed up the comprehension of the figure.

In Figure 7 and 9 it would be useful to indicate at the right side and at the top, respectively, the experimental condition (e.g., Figure 9, on the top of the left side ”forward” and right side “backward”).

In the figure legend the rankum test is mentioned. However, no description in the text of this analysis appears. Please provide descrption of the statistical analysis in the text.

Experimental design

Paradigm and Trial-Sequence. The authors should provide a more detailed description of the protocol.

The number of trials of the three protocols are the same. However, whether in P1 there are 12 target positions, in P2 and P3 only 2 endpoints appeared. This can affect the results in term of learning and transfer. Furthermore, the attentional load in the P1 with respect to P2 and P3 could be different. Please provide a comment.

Do target positions in P2 and P3 correspond to some of the target positions in P1? If so, it is not clear the need to perform P2 and P3.

The number of trials per each block and each target direction is not clearly described.

When looking at Table 1 and also to the figures it appears that protocols P2 and P3 are divided in P2a, P2b, P3a, P3b but they are not mentioned in the paradigm description.

Validity of the findings

Data Analysis

Lines 139-140. Why do the authors choose “the point in time 100 ms before the 45 percent of maximal hand-velocity” as movement onset? This sounds a little bit strange. Please motivate this choice or provide proper quotations.

Lines 155-159. Movement errors were fitted with linear and exponential functions, depending on the blocks. A statistical comparison among the parameters of the fits could be useful to compare the trend of initial movement error in the different protocols and directions.

Lines 160-168. The three protocols are balanced for total number of trials but not for movement direction. This implies that the number of repetitions per direction are very different and could have affected the statistical evaluation.

In the statistical analysis why the authors did not take into account the potential differences among the 12 movement directions?

Why in P1 “50 trials in the beginning and in the end of the block” were pooled together whether in P2 and P3 the trials were sorted by targets and only 15 movement were considered for the analysis?

The description of the tests used to provide the statistical results is missing.

Backward movements

The effects of the different protocols on movement performance are only discussed basing on the differences in the level of significance of the p value. In fact, the statistical evaluation is limited to comparison with-in the same group. A between subject analysis among the three groups of subjects is missing. Following the presented results it is difficult to conclude that the transfer of meaning occurred in a direction specific way. This is a strong limit of the work.

---

## Round 0.2 · Major Revisions

After reading the manuscript and the reviewers comments, my decision is to give you the possibility to address the raised issues.

Reviewer 2 ·

Basic reporting

The Figures are appropriate and improved from the first version of the manuscripts.

Experimental design

As requested, the authors provided a more detailed and satisfying description of the experimental protocol.

Validity of the findings

A description of the statistical evaluation of the data is still missing as also a between subject analysis among the groups. Since one of their conclusion concerns the comparison between groups (as stated by the authors in line 261), I think this lack remains a strong limit of the work. Why do they not directly inputs the data of paradigm 2 and 3 within the same statistical analysis?

---

## Author Rebuttal · Round 0.2

Dear Dr. Fadiga,

Thank you for your comments on our submitted manuscript. We carefully went through the comments from the two reviewers, and made a substantial amount of changes in the text and the figures. We believe the manuscript was greatly improved, and hope that you consider it to be fit for publication.

In particular, we reformulated our conclusions, to express more clearly that there are most likely more mechanisms involved in the learning of backward movements, but that our results strongly indicate that the directional-dependent transfer of learning from forward to backward movements is one of them. We also included an additional statistical test, directly comparing the differences of learning forward and backward movements between paradigms two and three (2 Targets – either opposite each other, or not), and a new Figure (Figure 10), that shows these differences for each subject separately.

In the following, please find our detailed replies to the reviewer's comments:

# Reviewer 1

*Instructions to subjects about movement speed*
> Included in Methods/Paradigms and Trial-Sequence (Lines 154-155)

*Report results of statistical analysis in text*
> Added in the text where the statistics of the results are discussed (198, 430)

*Figure with trajectories should be figure 3*
> The figure-order was changed accordingly

*Movement amplitued (5cm) is too small to speak of movements*
> 5cm indeed is a relatively small movement amplitude. However, we strongly feel that this amplitude is not too small to speak of real movements. First of all, in previous studies, for instance Krakauer et al. 2000, a similar movement magnitude has been used (6.2cm) to show visuomotor rotation learning and generalization over target direction (or rather: no generalization). Even more convincing are the typical movement trajectories we observed in our experiment. These exhibit a completely typical sequence, observed and described in many previous papers on motor learning: There is an initial phase in which subjects follow a movement direction planned prior to initiating the movement (demonstrated by the deviating direction under the visuomotor rotation), followed by the correcting movements induced by visual feedback (see Figure 3). Finally, movement velocities followed a typical bell-shaped curve. The features described are, in our mind, convincing enough to speak of real movements even given the relatively small movement amplitude.

*„Authors argue about movement direction as a major parameter of movement planning. In generally, this is true. However, in their experiment they manipulated only movement direction, movement amplitude was only 5cm, and they measured only direction errors. Thus, they must formulate more moderate conclusions.“*
> We agree with the reviewer that since we did not vary movement amplitude, we can not make any statement on the relative importance of direction vs. Amplitude in movement planning, and that some of our statements may have been formulated too strongly. We changed the respective passages in the text accordingly - especially the last paragraph (Lines 531-538), expressing more clearly that the transfer of learning movement direction is one, rather than the only mechanism.

*„They argued in favor of the 'vectorial planning hypothesis' because the significance lever, although present, was weaker in P3. Do they think that this is a sufficient argument? The last conclusion (lines 366-369) is not original."*

In addition to the comparison between significance levels between paradigms, we included a subject-wise test comparing the differences between forward and backward learning directly between paradigms (lines 216-219, lines 427-439). With this test, we could show that there is a statistically significant difference in the transfer of learning from forward to backward directions between the two paradigms, depending on the directions presented during forward movements (in the respective paradigms). Since this comparison is based on a small amount of data (subject-wise comparison), we also included a figure showing the values for each subject (Figure 10). The concluson mentioned by the reviewer was modified to state more clearly that we are specifically referring to backward movements (see lines 518-530).

## Reviewer 2

*Figures*

*„In all the figures I suggest to put a legend of the colors. This would simplify and speed up the comprehension of the figure."*

Done.

*In Figure 7 and 9 it would be useful to indicate at the right side and at the top, respectively, the experimental condition (e.g., Figure 9, on the top of the left side "forward" and right side "backward").*

Done.

*In the figure legend the rankum test is mentioned. However, no description in the text of this analysis appears. Please provide descrption of the statistical analysis in the text.*

The description was added at the appropriate locations (see line 198, 430).

*Experimental Design*
*Paradigm and Trial-Sequence. The authors should provide a more detailed description of the protocol.*

The experiment description was modified to include more detail and make it easier to follow (lines 101-164).

*The number of trials of the three protocols are the same. However, whether in P1 there are 12 target positions, in P2 and P3 only 2 endpoints appeared. This can affect the results in term of learning and transfer. Furthermore, the attentional load in the P1 with respect to P2 and P3 could be different. Please provide a comment.*

The reviewer is completely correct in observing the different number of repetitions in the different paradigms. We have added a clarifying comment on that fact in the methods section (Lines 138-143: 'It is important to note that since the target number differed between paradigms, but the trial number in each experimental block was kept constant, the number of repetitions to each target differed. In the first paradigm, each target was shown 10 times in the familiarization and washout blocks, and 30 times in the learning block. In the second and third paradigm, each target was presented 60 times during familiarization and washout and 180 times during learning.'
The attentional load was certainly different in the first paradigm presenting subjects with many more trials and having much less repetitions for each individual targets. Since our conclusions, however, are critically based on the comparisons of forward and backward movements within a

given paradigm and the comparison of paradigm 2 and 3 (in which equal numbers of repetitions were used), we do not think that the factor of repetitions is confounding our results. This point was also added in the text (Lines 291-296).

*Do target positions in P2 and P3 correspond to some of the target positions in P1? If so, it is not clear the need to perform P2 and P3.*

Indeed, all targets presented in the last two paradigms were also shown in the first. However, in paradigm one, subjects learned all directions during the forward movements, and, therefore, learning in backward movements could have been due to learning that direction during forward movements. To rule out this possibility, paradigm 3 was created, in which the movement directions of forward movements were clearly different from those required in backward movements. To rule out any confounding effects of the number of targets trained or the repetition numbers, additionally, paradigm 2 was introduced.

*The number of trials per each block and each target direction is not clearly described.*

The rewritten paradigm description states the target-number in each block and direction much more clearly than previously. (Lines 111-121)

*When looking at Table 1 and also to the figures it appears that protocols P2 and P3 are divided in P2a, P2b, P3a, P3b but they are not mentioned in the paradigm description.*

The labels a and b refer to the two different targets used in paradigms 2 and 3. This fact is now more clearly stated in the paradigm description (lines 131-137) and also added in Figure 1.

*Data Analysis*
*Lines 139-140. Why do the authors choose "the point in time 100 ms before the 45 percent of maximal hand-velocity" as movement onset? This sounds a little bit strange. Please motivate this choice or provide proper quotations.*

The parameters in this procedure were found empirically, by comparing the movement onsets found automatically with the speed profiles and trajectories of the movements in all individual trials. The procedure was relatively stable to small changes in the parameters, so the exact numbers (100ms and 45 percent, instead of e.g. 90ms and 40 percent) are somewhat arbitrary. The important factors in determining the movement onset were (1) finding a point in time at which the hand had not yet moved substantially from the starting location and (2) finding a movement onset that is not just a quiver – or a false start (in both cases, the initial movement direction from that point would not be task-related!). The text has been changed to clarify this point (lines 173-183).

*Lines 155-159. Movement errors were fitted with linear and exponential functions, depending on the blocks. A statistical comparison among the parameters of the fits could be useful to compare the trend of initial movement error in the different protocols and directions.*

The exponential and linear fits to the data are added mainly for visualization purposes. Although the offsets (parameter a) indeed follow the same pattern as the comparison of the pooled errors in the end of learning (i.e. Errors in forward and backward movements are closer together in the end of the learning in paradigm 1 and 3 than in paradigm 3), we found it problematic to make a statistical comparison between the fitted parameters, since they represent single numbers based on the data from all subjects. Data from single subjects is highly variable, so doing the fit for single subjects, and then compare  the parameters between experimental groups was no feasible option. Even though we therefore do not present a statistical comparison of the fit parameters, we felt that it would be helpful for the reader to show the parameters and errors of the fits.

*Lines 160-168. The three protocols are balanced for total number of trials but not for movement direction. This implies that the number of repetitions per direction are very different and could*

*have affected the statistical evaluation.*

See also the second point regarding experimental design (Lines 138-143: 'It is important to note that since the target number differed between paradigms, but the trial number in each experimental block was kept constant, the number of repetitions to each target differed. In the first paradigm, each target was shown 10 times in the familiarization and washout blocks, and 30 times in the learning block. In the second and third paradigm, each target was presented 60 times during familiarization and washout and 180 times during learning.'). The main focus of statistical comparison is on the last two paradigms, in which the numbers of targets and repetitions is equal.

*In the statistical analysis why the authors did not take into account the potential differences among the 12 movement directions?*

We added some comments on this in the text: see Lines 204-209 'For the first paradigm, we pooled over the first and last 50 trials, irrespective of target direction. Target direction was ignored in this paradigm because we expected a high degree of generalization of learning between nearby targets (also see Discussion). For the second and third paradigm, we first sorted trials by target direction and then pooled over the first and last 15 trials separately for each target (P2a and P2b in the second paradigm; P3a and P3b in the third paradigm).' and Lines 273-282 'For the presentation of initial movement errors in the first paradigm, we ignored target directions when looking at the time-course of learning (see Figure 5). Due to the pseudo-random presentation of many target directions, consecutive trials to the same target can be separated by many trials to other targets. If targets are learned completely separately, this should not strongly affect the time-course of learning the rotation for single targets (comparesee Krakauer et al. 2000), but in our first paradigm, targets were quite close together (30 degree) so we expected to see at least some generalization between nearby target locations. Instead of trying to correct for this during the analysis (e.g. by taking into account the presentation order of targets), we decided to look at consecutive trials irrespective of target direction for the paradigm with 12 targets.'

*Why in P1 "50 trials in the beginning and in the end of the block" were pooled together whether in P2 and P3 the trials were sorted by targets and only 15 movement were considered for the analysis?*

The idea of pooling was to reduce trial-by-trial variability, but at the same time take as little trials as possible since we wanted to show the performance in trials before the rotation is (completely) learned. In P1, 50 trials were chosen, since we also wanted to have a fairly even distribution of target-repetitions in the sample (50 trials ~ 4 repetitions per target).

*The description of the tests used to provide the statistical results is missing.*

added in the text (line 198, 430)

*Conclusions*
*The effects of the different protocols on movement performance are only discussed basing on the differences in the level of significance of the p value. In fact, the statistical evaluation is limited to comparison with-in the same group. A between subject analysis among the three groups of subjects is missing. Following the presented results it is difficult to conclude that the transfer of meaning occurred in a direction specific way. This is a strong limit of the work.*

(same as the last point of Reviewer 1) Instead of the comparison between significance levels between paradigms, we included a between-subject test, directly comparing the differences between forward and backward learning directly between paradigms. This test shows that this difference is significantly larger in the paradigm where directions between forward and backward movements do not match (paradigm 2) – even when only picking movements to the target that was presented in both paradigms. The most likely explanation for this difference between the experimental groups is a direction-specific transfer of learning, which affects backward movements in the third, but not in the second paradigm. (Methods, Lines 216-219, Results and Discussion, Lines 427-439).

Since the between-subject comparison is based on a small amount of data (the number of subjects in each group), we also included a figure showing the values for each subject separately (Figure 10).

---

## Round 0.3 · accepted · Accept

The raised issues have been addressed in a satisfactory way.